# Multiple Criteria Optimization (MCO): A gene selection deterministic tool in RStudio

Isis Narváez-Bandera[1], Deiver Suárez-Gómez[1], Clara E. Isaza[1,2,3,4], Mauricio Cabrera-Ríos[1,5]*

1 Bioengineering Graduate Program, The Applied Optimization Group, University of Puerto Rico-Mayagüez, Mayagüez, Puerto Rico, 2 Public Health Program, Ponce Health Sciences University, Ponce, Puerto Rico, 3 Basic Sciences Department, Ponce Health Sciences University, Ponce, Puerto Rico, 4 Biology Department, University of Puerto Rico-Mayagüez, Mayagüez, Puerto Rico, 5 Industrial Engineering Department, The Applied Optimization Group, University of Puerto Rico-Mayagüez, Mayagüez, Puerto Rico

* mauricio.cabrera1@upr.edu

**Data Availability Statement:** All four microarray datasets relating to Parkinson's Disease (PD) are available from the Gene Expression Omnibus (GEO) repository (http://www.ncbi.nlm.nih.gov/

## Abstract

Identifying genes with the largest expression changes (gene selection) to characterize a given condition is a popular first step to drive exploration into molecular mechanisms and is, therefore, paramount for therapeutic development. Reproducibility in the sciences makes it necessary to emphasize objectivity and systematic repeatability in biological and informatics analyses, including gene selection. With these two characteristics in mind, in previous works our research team has proposed using multiple criteria optimization (MCO) in gene selection to analyze microarray datasets. The result of this effort is the MCO algorithm, which selects genes with the largest expression changes without user manipulation of neither informatics nor statistical parameters. Furthermore, the user is not required to choose either a preference structure among multiple measures or a predetermined quantity of genes to be deemed significant a priori. This implies that using the same datasets and performance measures (PMs), the method will converge to the same set of selected differentially expressed genes (repeatability) despite who carries out the analysis (objectivity). The present work describes the development of an open-source tool in RStudio to enable both: (1) individual analysis of single datasets with two or three PMs and (2) meta-analysis with up to five microarray datasets, using one PM from each dataset. The capabilities afforded by the code include license-free portability and the possibility to carry out analyses via modest computer hardware, such as personal laptops. The code provides affordable, repeatable, and objective detection of differentially expressed genes from microarrays. It can be used to analyze other experiments with similar experimental comparative layouts, such as micro-RNA arrays and protein arrays, among others. As a demonstration of the capabilities of the code, the analysis of four publicly-available microarray datasets related to Parkinson´s Disease (PD) is presented here, treating each dataset individually or as a four-way meta-analysis. These MCO-supported analyses made it possible to identify MMP9 and TUBB2A as potential PD genetic biomarkers based on their persistent appearance across each of the case studies. A literature search confirmed the importance of these genes in PD and indeed as PD biomarkers, which evidences the code´s potential.

geo). Accession numbers: GSE99039, GSE18838, GSE19587, and GSE57475.

**Funding:** The project described was supported by Award Number U54MD007579 from the National Institute on Minority Health and Health Disparities. The content is solely the responsibility of the authors and does not necessarily represent the official views of the National Institutes of Health.

**Competing interests:** The authors have declared that no competing interests exist.

# Introduction

The analysis of gene expression leads to insight into biological processes and identification of biomarkers, as well as characterization of differing responses to therapy by individuals. Some of the first high throughput experiments used to analyze gene expression were microarray experiments. These were and still are typically used in comparative experiments between case and control groups to identify differentially expressed genes. The data generated by experiments like this populate large public repositories such as Gene Expression Omnibus. Ideally, it would be possible to pool several of these experiments to carry out a meta-analysis and arrive at statistically more robust conclusions about potential biomarkers. However, the fact that many of these experiments are measured in different units and scales has often made simultaneous analysis difficult. Nevertheless, several techniques are used to discover biomarkers using microarrays [1, 2]; these range from traditional statistical models to more computationally complex machine learning methods. For instance, in [3–6], methods based on genetic algorithms, spearman correlation, relief-F, and joint mutual information, among others, are used to analyze microarray data with that purpose. However, the outputs of these methods, require the configuration of a varying number of parameters that significantly affect their results. This hampers both analysis objectivity and repeatability. To this end, our research group in Camacho et al [7] proposed the MCO approach as it appears in this manuscript, which can analyze microarrays and other -omics datasets relying on Pareto-optimality conditions. The MCO-based analysis is carried out without the assumption of underlying statistical distributions a priori, the selection of a threshold value, or the adjustment of parameters by the user. Moreover, MCO presents a ranking based on the simultaneous consideration of performance measures included in the analysis. Our succeeding works presented in Cruz-Rivera et al. [8], Lorenzo et al. [9], Isaza et al. [10], successfully developed and applied MCO for gene selection in Alzheimer's Disease, cervix cancer, and lung cancer, respectively. In the present work, MCO is fully automated using R. The resulting code maintains the nonparametric qualities of MCO and minimizes possible errors due to manual handling of data. Lead time to carry out analysis is also significantly decreased, making MCO a convenient and powerful tool to support the search for potential biomarkers. The MCO R tool can be accessed from the address https://server-deiver.shinyapps.io/MCO_TURBO/.

# Design and implementation

## MCO algorithm

As discussed in this work, the MCO algorithm requires at least one treatment vs control, comparative, high-throughput, replicated experiment. Several microarray datasets follow this organization in the characterization of relative gene expression. The comparative layout allows obtaining PMs for every gene in the experiment to measure differences in relative expression. For example, one PM can be the absolute value of the difference of means between the two groups (treatment, control), another can be the absolute value of the difference of medians between them, and so on. It is doubtful that selecting genes solely using one PM at a time will coincide precisely with the resulting selection. This evidences that there exists conflicts between different PMs. In the case of MCO, each gene is represented through several PMs -very much like a coordinate system- and the final selection is made up of the genes showing the best possible compromises between all PMs. In mathematical terms, this refers to identifying the Pareto-efficient solutions of the associated multiple criteria optimization problem. The Pareto-efficient solutions, in turn, form the Pareto-efficient frontier of such problem, from here on also referred to simply as the frontier. There exists sufficiency in Pareto-optimality,

which means that for the genes that meet the Pareto-efficient conditions (i.e., those lying in the frontier), no other gene can be found in the experiment offering a better compromise between the PMs at hand. This confers certainty to the results.

## MCO tool

The present work describes the development of an open-source code in RStudio of the MCO algorithm. The tool permits to identify, in a single run, the first F frontiers in an analysis. The user can specify F to create a hierarchy of genes organized in succeeding frontiers. The MCO tool was designed to support: (1) individual analysis of microarray datasets using two or three PMs, and (2) meta-analysis using two to five different datasets with one PM from each dataset (see Fig 1). It should be noted that the MCO algorithm could handle more datasets in a meta-analysis; however, to keep the computational cost low, the MCO tool is limited to a maximum of five data-sets. The application of MCO results in sets of genes organized in $F$ frontiers with decreasing levels of significance. In both options, the MCO algorithm follows three steps (Algorithm 1). In the first step, the PMs are selected from a predefined list, including the median, the mean, the mode, the third quartile, or a quantile of interest to the analyst. When the analysis is performed for an individual dataset, choosing between two or three PMs is possible. In the case of a meta-analysis, the default PM is the median from each dataset. The difference of the selected PMs between cases and controls is calculated for each gene. The absolute value function is further applied to then be subjected to a linear transformation aimed to set up a minimization MCO problem. In the second step, the MCO tool allows the user to divide the dataset into $S$ groups (parallelism) to address RAM constraints when using hardware with modest capabilities. The MCO tool proceeds to find the local frontier of each group. The genes in each one of the $S$ local frontiers are analyzed together to find the global Pareto-efficient frontier. This second step returns the genes with the best possible balances among the PMs to be minimized and are the ones identified as potential biomarkers. The third step entails applying the MCO algorithm $F$ times, each time removing the previous frontier. This approach returns genes organized hierarchically in $F$ frontiers.

**Algorithm 1:** Pseudocode of the MCO tool, the procedure to carry out the selection of the first F Pareto-efficient frontiers. $D_k$ represents the K datasets to be analyzed, in this implementation K falls between 1 and 5. PMc is the number of performance measures used to quantify relative expression changes between treatment and control. F is the number of successive frontiers to be presented in the analysis. The output is presented as a list of genes, hierarchically organized by frontier number.

```
Input:
Dₖ ← Number of datasets |Dₖ ← (Xₘ, Yₙ) |k ∈ 1, 2, ..., K;
     Xₘ ∈ samples- m = 1, 2, ..., M;
     Yₙ ∈ gene sets- n = 1, 2, ..., N;
PMᴄ ← Number of PMs |c = 1, 2, ...C;
     (c = PMs in conflict);
S ← Number of groups in which the genes are split |s = 1, 2, ..., S;
F ← Number of frontiers |F = 1, 2, ..., f;
for Dₖ ← 1 to K do
  for PM ← 1 to c do
    for s ← 1 to S do
      for f ← 1 to F do
        MCO function (PMᴄ)
      end
    end
  end
end
Output: Genes sets for each frontiers F
```

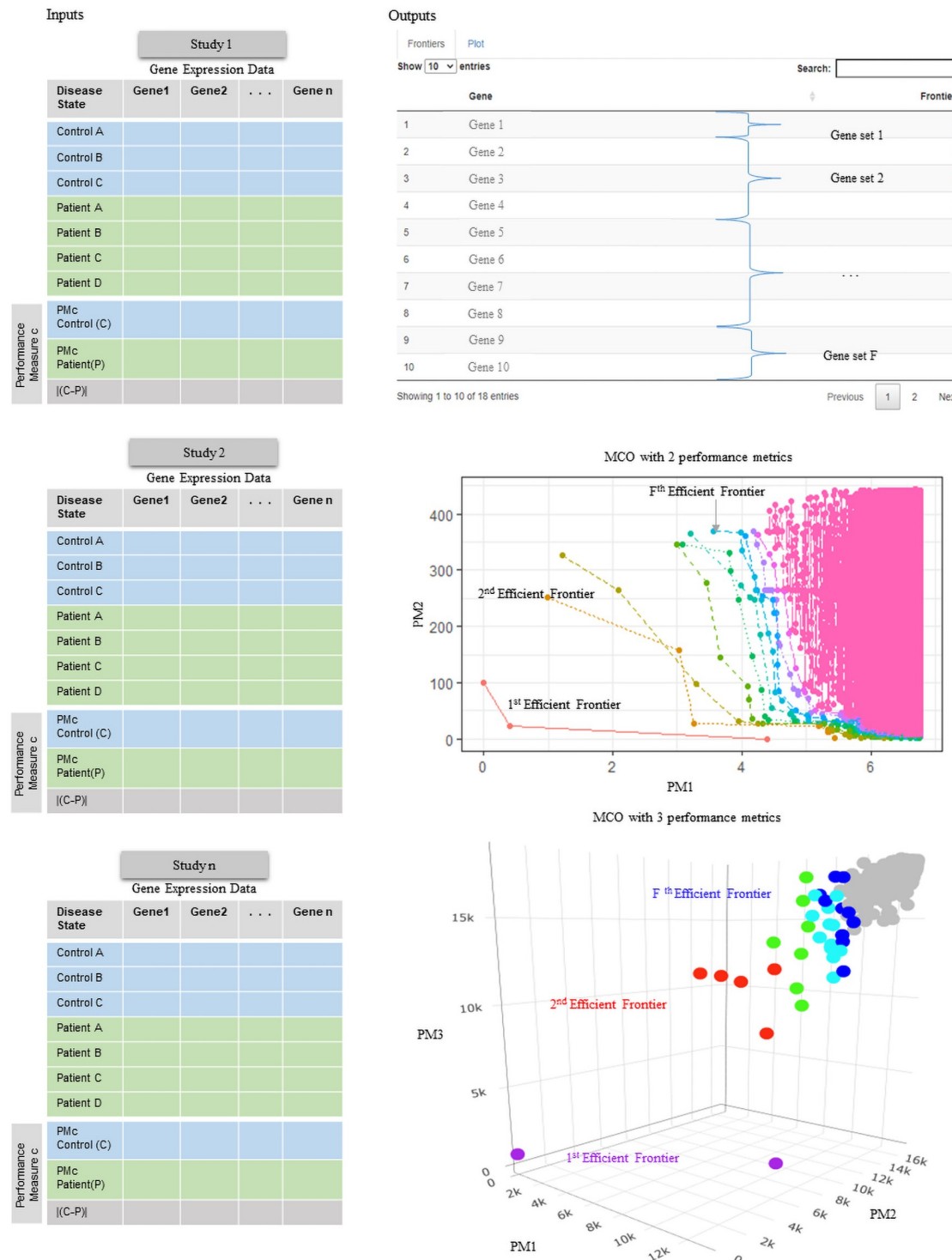

**Fig 1. Overview of the MCO tool.** Gene expression datasets can be analyzed individually or by combining several datasets. The PMs can be generated for individual analysis or meta-analysis of up to five datasets simultaneously. The MCO results can then be visualized by the function plotMCO.

**Table 1. List of PD studies from GEO.**

| GEO accession | Year | Platform | Probe sets | Genes | Control (Male/Fem) | Condition (Male/Fem) |
|---|---|---|---|---|---|---|
| GSE99039 | 2017 | GPL570 | 54675 | 23520 | 212 (70/142) | 191 (101/90) |
| GSE18838 | 2010 | GPL5175 | 316919 | 17326 | 11 (6/5) | 17 (13/4) |
| GSE19587 | 2010 | GPL571 | 22277 | 13515 | 10 (6/4) | 12 (6/6) |
| GSE57475 | 2015 | GPL6947 | 49576 | 25146 | 49 (26/23) | 93 (62/31) |

## Results

### MCO tool—Application

To demostrate how the MCO tool works, four microarray datasets relating to Parkinson's Disease (PD) were selected from the Gene Expression Omnibus (GEO) repository. These are GSE99039 [11], GSE18838 [12], GSE19587 [13], and GSE57475 [14] (http://www.ncbi.nlm.nih.gov/geo). In the aggregation of the four datasets, there are 282 control samples (108 male/174 female) and 313 PD samples (182 male/131 female). Table 1 lists their characteristics. These datasets were selected using the following query filters: (1) ´Parkinson's´ was used as a keyword, (2) the type of dataset was defined as 'expression profiling by array', (3) the organism was 'Homo sapiens', and (4) the gender information was set to 'available'. The latter was included since it has been considered relevant to differentiate PD profiles [15]. Following these criteria, each dataset contained four distinct groups arising from the intersection of sex and type of sample: MaleControl, MalePD, FemaleControl, and FemalePD.

Two types of analysis are presented: individual analysis of datasets and meta-analysis of multiple datasets. For the first type, we treated each of the four datasets individually. For the second type, we meta-analyzed three datasets simultaneously and four datasets simultaneously. Six analysis instances result in this manner: four individual analyses and two meta-analyses. In each instance, the goal was to detect genes with significant relative expression changes through MCO to characterize and infer their biological meaning in PD. MCO requires that at least two PMs are identified to work. On each of the four individual analysis instances, two PMs were used: the absolute value of the difference of the means and the absolute value of the difference of the medians between the pair of groups under comparison. The MCO analyses in these instances were, then, all 2-dimensional (2D). On the other hand, in the meta-analysis involving three datasets, there were three PMs: the absolute value of the difference of medians between the pair of groups under comparison from each dataset. The MCO analyses involved here are 3D. Finally, in the last meta-analysis instance involving four datasets, there were four PMs: the absolute value of the difference of medians between the pair of groups under comparison from each dataset, resulting in an MCO analysis that is 4D. On each of the six analysis instances, MCO was applied four times (Fig 2A): MCO 1 compares the pair of groups MaleControl—MalePD; MCO 2 compares the groups FemaleControl—FemalePD; MCO 3 compares the groups MaleControl—FemaleControl; and MCO 4, the groups MalePD—FemalePD. The genes of interest were those with biomarking potential solely for PD. Following set theory -see Venn diagram in Fig 2B this results in selecting the genes in the intersection of MCO 1 and MCO 2 that are not in MCO 3 or MCO 4.

One last difference is that, in the four individual analysis instances, the number of frontiers, F, was set to 10. In the two meta-analyses, it was set to 1 to keep results manageable. The results of the six analysis instances can be identified by defining: number of datasets in the analysis, the number of PMs (dimensions), and the last four digits of the identifiers of the datasets involved. Explicitly, the instances are: 1–2D-9039, 1–2D-8838, 1–2D-9587, 1–2D-7475, 3–3D-9039/8838/9587, 4–4D-9039/8838/9587/7475. The genes of interest for the four individual analysis instances can be consulted in Table 2. For illustration purposes, Fig 3 shows the

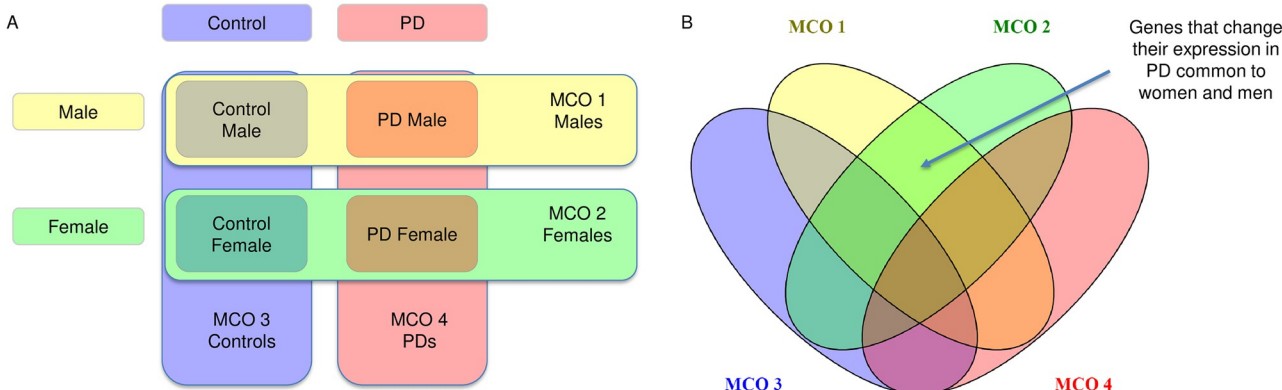

**Fig 2. The general scheme for all six analysis instances.** (A) MCO 1 compares the pair of groups MaleControl—FemaleControl; MCO 2 compares the groups FemaleControl—FemalePD; MCO 3 compares the groups MaleControl—FemaleControl; and MCO 4, the groups MalePD-FemalePD; (B) The genes of interest were those with biomarking potential solely for PD. Following set theory, this selects the genes in the intersection of MCO 1 and MCO 2 that are not in MCO 3 or MCO 4.

**Table 2. Genes of interest from individual analysis instances 1–2D-9039, 1–2D-8838, 1–2D-9587, and 1–2D-7457 and the references supporting their roles in Parkinson's Disease (PD) or neurodegenerative diseases (ND).**

| Individual Analysis Instance | Gene symbol | Gene name | Reference related to | |
|---|---|---|---|---|
| | | | PD | ND |
| 1–2D-9039 | TUBB2A | Tubulin Beta 2A Class IIa | [16–18] | [19] |
| | CFD | Complement Factor D | [20] | [21] |
| | PTGDS | Prostaglandin D2 Synthase | [16] | |
| | LRRN3 | Leucine Rich Repeat Neuronal 3 | [16, 22] | |
| | ANXA3 | Annexin A3 | | [23, 24] |
| | GPR97 | G Protein-Coupled Receptor 97 | [25] | |
| | PRKCD | Protein Kinase C Delta | [26, 27] | |
| | MMP9 | Matrix Metallopeptidase 9 | [28–30] | |
| | PGLYRP1 | Peptidoglycan Recognition Protein 1 | [31] | |
| | SPI1 | Spi-1 Proto-Oncogene | [22] | |
| 1–2D-8838 | ND6 | NADH Dehydrogenase Subunit 6 | [32] | |
| | GTF2B | General Transcription Factor IIB | | |
| | RPL18 | Ribosomal Protein L18 | | |
| | TAGLN2 | Transgelin 2 | | |
| | TMEM14B | Transmembrane Protein 14B | | |
| | GABARAP | GABA(A) Receptor-Associated Protein | [18] | |
| | HIST1H1E | Histone Cluster 1 | | |
| 1–2D-9587 | OPHN1 | Oligophrenin 1 | | |
| 1–2D-7475 | OAZ1 | Ornithine Decarboxylase Antizyme 1 | | |
| | EEF1A1 | Elongation Factor 1-Alpha 1 | [33, 34] | |
| | ARHGDIB | Rho GDP Dissociation Inhibitor Beta | [35] | |
| | HBD | Hemoglobin Subunit Delta | [36] | |
| | CFD | Complement Factor D | [20] | [21] |
| | UBA52 | Ubiquitin Carboxyl Extension Protein 52 | [18] | |

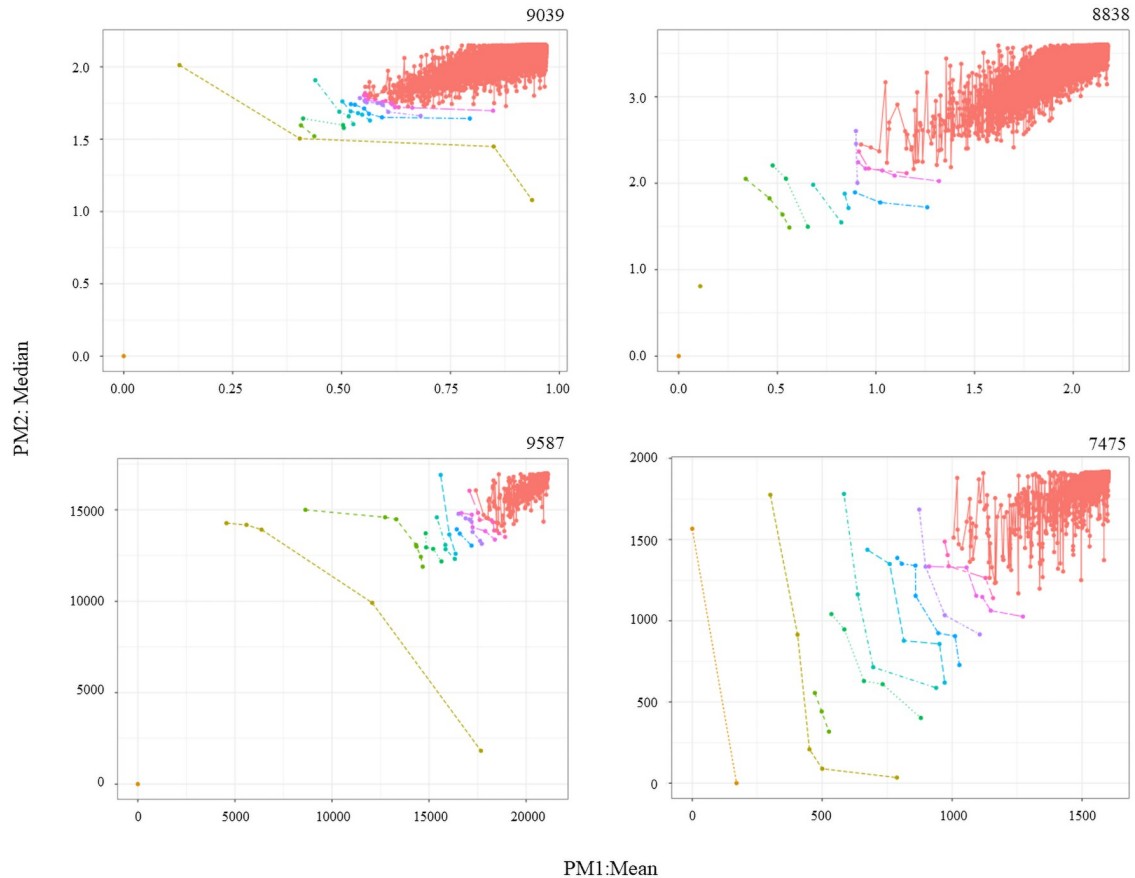

**Fig 3. MCO 1: MalesPD Vs MaleControl.** Graphical representations for MCO 1 associated to individual analysis instances 1–2D-9039, 1–2D-8838, 1–2D-9587, and 1–2D-7457. Solutions toward the origin (0,0) are more significant.

graphical results for MCO 1 on each dataset, while Fig 4 does the same for MCO 2. The complete lists identifying ten frontiers on each dataset can be found in the S2-S5 Tables in S1 File.

Table 3 contains the genes of interest for the two meta-analysis instances. In 3–3D-9039/8838/9587, four genes were identified as potential biomarkers: MMP9, RPS11, TUBA1B, and TUBB2A. Fig 5 shows the results for MCO 1 and MCO 2 for illustration purposes for this instance. The complete lists for MCO 1 through MCO 3 can be found in the S6 Table in S1 File.

In 4–4D-9039/8838/9587/7475, 10 genes of interest were identified: EEF2, MMP9, CFD, DAZAP2, MYL6, ARHGDIB, RPL18, RPS11, CD81, and TUBB2A, as shown in Table 3. The complete lists for MCO 1 through MCO 4 can be found in the S7 Table in S1 File.

The Venn diagrams in Fig 6 show the overlap within the meta-analysis instances 3–3D-9039/8838/9587 in (A) and 4–4D-9039/8838/9587/7475 in (B). Notably, three genes (MMP9, RPS11, and TUBB2A) were identified in both instances. In addition, two out of these three, MMP9 and TUBB2A, were also identified in instance 1–2D-9039.

## Discussion

In the literature, our team performed a series of queries combining the name of each of the ten genes and either 'Parkinson´s Disease' or 'Neurodegenerative' to look for related biological

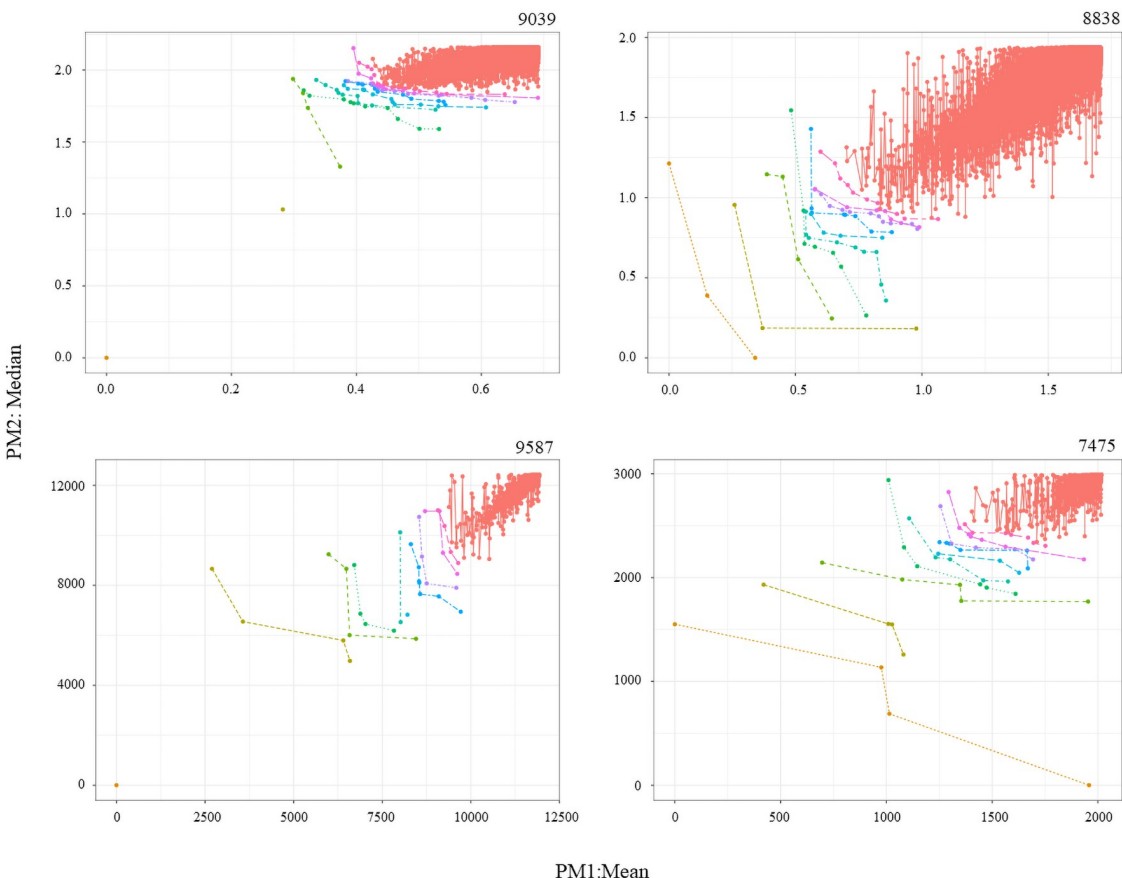

**Fig 4. MCO 2: FemalesPD Vs FemaleControl.** Graphical representations for MCO 2 associated to individual analysis instances 1–2D-9039, 1–2D-8838, 1–2D-9587, and 1–2D-7457. Solutions toward the origin (0,0) are more significant.

**Table 3. Genes of interest for meta-analysis instances 3–3D-9039/8838/9587 and 4–4D-9039/8838/9587/7475 and the references supporting their roles in Parkinson's Disease (PD) or neurodegenerative diseases (ND).**

| Meta-Analysis Instance | Gene symbol | Gene name | Reference related to | |
|---|---|---|---|---|
| | | | PD | ND |
| 3–3D -9039 /8838 /9587 | MMP9 | Matrix Metallopeptidase 9 | [28–30] | |
| | RPS11 | Ribosomal Protein S11 | [37] | |
| | TUBA1B | Tubulin Alpha 1b, K-ALPHA-1 | [35] | |
| | TUBB2A | Tubulin Beta 2A Class IIa | [16–18] | [19] |
| 4–4D -9039 /8838 /9587 /7475 | EEF2 | Eukaryotic Translation Elongation Factor 2 | [38, 39] | |
| | MMP9 | Matrix Metallopeptidase 9 | [28–30] | |
| | CFD | Complement Factor D | [20] | [21] |
| | DAZAP2 | DAZ Associated Protein 2 | | |
| | MYL6 | Myosin Light Chain 6 | [40] | |
| | ARHGDIB | Rho GDP Dissociation Inhibitor Beta | [35] | |
| | RPL18 | Ribosomal Protein L18 | | |
| | RPS11 | Ribosomal Protein S11 | [37] | |
| | CD81 | CD81 Molecule | [41] | |
| | TUBB2A | Tubulin Beta 2A Class IIa | [16–18] | [19] |

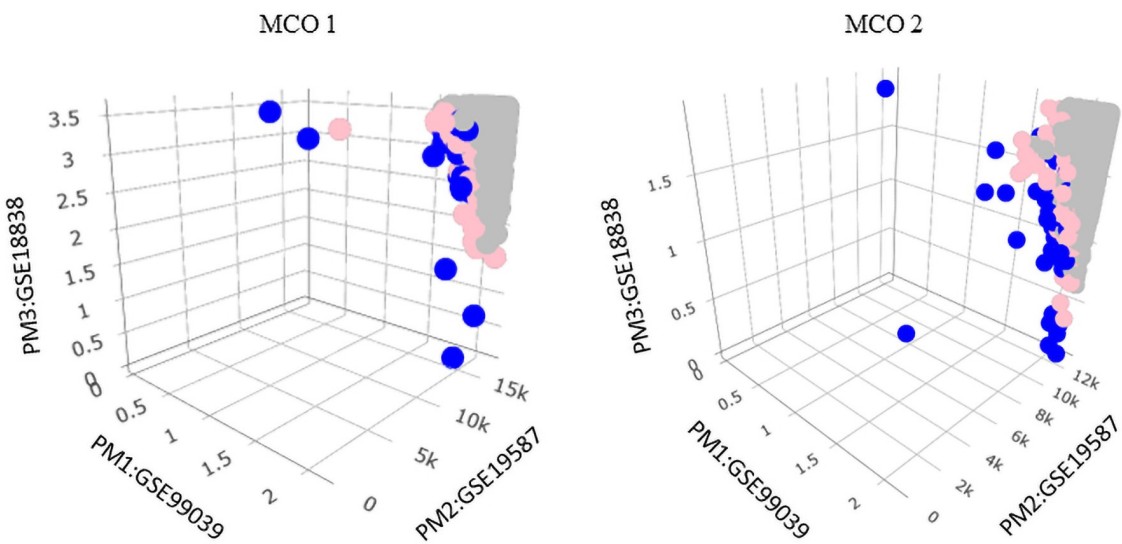

**Fig 5. Meta-analysis instance 3–3D-9039/8838/9587.** Graphical representations for MCO 1 and MCO 2 are associated with this instance. Solutions toward the origin (0,0,0) are more significant.

evidence. This process found out that 9 out of 10 genes (MMP9, RPS11, TUBB2A, EEF2, CFD, DAZAP2, MYL6, ARHGDIB, RPL18, and CD81) identified in the meta-analysis instances have appeared in the literature as related to PD or neurodegenerative conditions. For instance, MMP9, a protein-coding gene, appears in 17 articles describing a direct association with PD. In [30], MMP9 was identified as a potential marker for Lewy body disorders, i.e. Parkinson's Disease. In [16], using the LASSO algorithm with 10-fold cross-validation cycles, TUBB2A was one of the 25 genes selected as differentially expressed mRNAs for PD. Furthermore, in [17], it is argued that TUBB2A is a molecular biomarker for PD in the blood, supporting

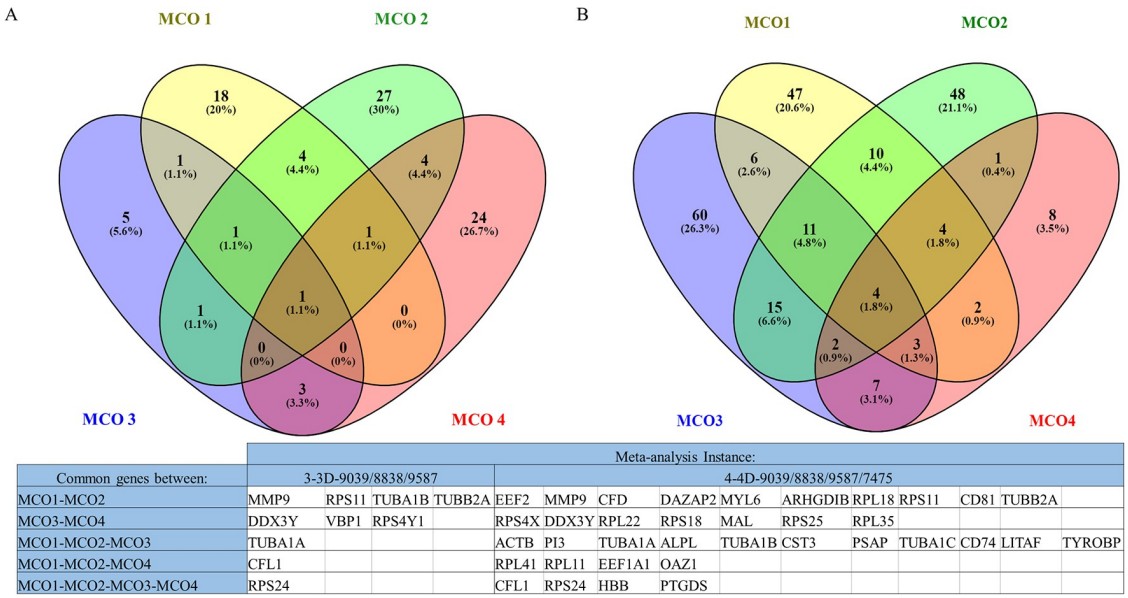

| Common genes between: | Meta-analysis Instance: | | | | | | | | | | | |
|---|---|---|---|---|---|---|---|---|---|---|---|---|
| | 3-3D-9039/8838/9587 | | | | 4-4D-9039/8838/9587/7475 | | | | | | | |
| MCO1-MCO2 | MMP9 | RPS11 | TUBA1B | TUBB2A | EEF2 | MMP9 | CFD | DAZAP2 | MYL6 | ARHGDIB | RPL18 | RPS11 CD81 TUBB2A |
| MCO3-MCO4 | DDX3Y | VBP1 | RPS4Y1 | | RPS4X | DDX3Y | RPL22 | RPS18 | MAL | RPS25 | RPL35 | |
| MCO1-MCO2-MCO3 | TUBA1A | | | | ACTB | PI3 | TUBA1A | ALPL | TUBA1B | CST3 | PSAP | TUBA1C CD74 LITAF TYROBP |
| MCO1-MCO2-MCO4 | CFL1 | | | | RPL41 | RPL11 | EEF1A1 | OAZ1 | | | | |
| MCO1-MCO2-MCO3-MCO4 | RPS24 | | | | CFL1 | RPS24 | HBB | PTGDS | | | | |

**Fig 6. Venn diagrams of meta-analysis instances 3–3D-9039/8838/9587 in (A) and 4–4D-9039/8838/9587/7475 in (B).** The accompanying table lists the genes in each intersection.

**Table 4. Pathways enriched for the genes of interest identified in meta-analysis.**

| ID | Description | GeneRatio | BgRatio | pvalue | p.adjusted | qvalue | Gene | Count |
|---|---|---|---|---|---|---|---|---|
| R-HSA-156902 | Peptide chain elongation | 3/9 | 89/10856 | 4.32E-05 | 0.003275522 | 0.001607 | EEF2/ RPL18/ RPS11 | 3 |
| R-HSA-156842 | Eukaryotic Translation Elongation | 3/9 | 93/10856 | 4.93E-05 | 0.003275522 | 0.001607 | EEF2/ RPL18/ RPS11 | 3 |
| R-HSA-166658 | Complement cascade | 2/9 | 58/10856 | 9.86E-04 | 0.028553514 | 0.014011 | CFD/ CD81 | 2 |
| R-HSA-2766 | Translation | 3/9 | 291/10856 | 1.42E-03 | 0.028553514 | 0.014011 | EEF2/ RPL18/ RPS11 | 3 |
| R-HSA-192823 | Viral mRNA Translation | 2/9 | 89/10856 | 2.30E-03 | 0.028553514 | 0.014011 | RPL18/ RPS11 | 2 |
| R-HSA-2682334 | EPH-Ephrin signaling | 2/9 | 92/10856 | 2.46E-03 | 0.028553514 | 0.014011 | MMP9/ MYL6 | 2 |

similar assertions in [18]. The authors in [17] demonstrated that TUBB2A in reduced expression reasonably predicted PD as a blood biomarker via a meta-analysis of 11 datasets from GEO (8 from substantia nigra and 3 from blood samples) with further validation analyzing mRNA expression levels in the blood of 50 sporadic PD patients and 50 control subjects. In agreement with known biology, TUBB2A was one of the top identified genes from both meta-analysis instances (Table 3). TUBB2A is a strong candidate for more in-depth explorations at the experimental level for PD. The fact that MMP9 and TUBB2A appeared in the results of instances 3–3D-9039/8838/9587 and 4–4D-9039/8838/9587/7475 as well as in 1–2D-9039, supports the sensitivity of our method to detect genes that play a crucial role in the disease under study. Besides this supporting biological evidence on the efficacy of the MCO tool, the code is also computationally efficient as it can meta-analyze up to five datasets simultaneously. The largest instance presented here, 4–4D-9039/8838/9587/7475, took around 10 minutes to process for any of the MCO 1 thru MCO 4 analyses on a personal laptop with a 2.90 GHz Intel Core i7 CPU and 16G RAM.

The ten genes of interest identified in meta-analysis instance 4–4D-9039/8838/9587/7475 were the subject of gene ontology (GO) enrichment analysis using ReactomePA R package [42] and Enrichr tool [43]. Table 4 lists the results from enrichPathway, and the Table 5 the results from Enrichr.

**Table 5. GO biological process.**

| Term | P-value | Overlap_genes |
|---|---|---|
| Translation (GO:0006412) | 0.000137136 | [EEF2, RPL18, RPS11] |
| Cellular macromolecule biosynthetic process (GO:0034645) | 0.000423823 | [EEF2, RPL18, RPS11] |
| SRP-dependent cotranslational protein targeting to membrane (GO:0006614) | 0.000880239 | [RPL18, RPS11] |
| Cytoplasmic translation (GO:0002181) | 0.000939488 | [RPL18, RPS11] |
| Cotranslational protein targeting to membrane (GO:0006613) | 0.000959656 | [RPL18, RPS11] |
| Protein targeting to ER (GO:0045047) | 0.001150535 | [RPL18, RPS11] |
| Nuclear-transcribed mRNA catabolic process, nonsense-mediated decay (GO:0000184) | 0.001382294 | [RPL18, RPS11] |
| neutrophil degranulation (GO:0043312) | 0.001462415 | [CFD, EEF2, MMP9] |
| neutrophil activation involved in immune response (GO:0002283) | 0.001497690 | [CFD, EEF2, MMP9] |
| neutrophil mediated immunity (GO:0002446) | 0.001524499 | [CFD, EEF2, MMP9] |

Having a short solution list made it possible to perform an in-depth literature search for each gene. The information added to the ontology analyses and found in which pathways the unlisted genes could be included. From the solution list: EEF2, RPL18, RPS11 code for proteins involved in protein synthesis (translation). It has been reported in various studies that protein synthesis is affected in PD and that some ribosomal proteins have expression changes in the condition [38, 44, 45], and eEF2 (the protein product of EEF2) has been reported to be expressed less in PD [45]. The DAZAP2 gene product, not included in the ontology analysis, could also be associated with translation (protein synthesis). The association is through the DAZAP2 gene product role in stress granules (SGs) that enclose different translation system components when the cell is under stress [46]. The DAZAP2 gene product also participates in translation through interactions with eukaryotic initiation factor 4G (https://www.genecards.org/cgi-bin/carddisp.pl?gene=DAZAP2). The ARHGDIB and TUBB2A gene products have roles in cytoskeletal organization and dynamics. The expression of genes for proteins involved in cytoskeleton dynamics, such as tubulin, changes in PD [44]. MMP9 and CD81 gene products are involved in cell motility and extracellular matrix dynamics, MMP9 expression changes in PD, and amyotrophic lateral sclerosis [47].

## Supporting information

**S1 File. Contains a comparative study of MCO Vs CFS, IG and eBayes gene selection methods and all the supporting tables and figures.**
(ZIP)

## Author Contributions

**Conceptualization:** Clara E. Isaza, Mauricio Cabrera-Ríos.

**Data curation:** Isis Narváez-Bandera, Deiver Suárez-Gómez.

**Formal analysis:** Isis Narváez-Bandera.

**Funding acquisition:** Clara E. Isaza.

**Investigation:** Isis Narváez-Bandera, Deiver Suárez-Gómez, Mauricio Cabrera-Ríos.

**Methodology:** Isis Narváez-Bandera.

**Project administration:** Clara E. Isaza, Mauricio Cabrera-Ríos.

**Resources:** Mauricio Cabrera-Ríos.

**Software:** Isis Narváez-Bandera, Deiver Suárez-Gómez.

**Supervision:** Clara E. Isaza, Mauricio Cabrera-Ríos.

**Validation:** Isis Narváez-Bandera, Deiver Suárez-Gómez.

**Visualization:** Isis Narváez-Bandera, Deiver Suárez-Gómez.

**Writing – original draft:** Isis Narváez-Bandera.

**Writing – review & editing:** Isis Narváez-Bandera, Clara E. Isaza, Mauricio Cabrera-Ríos.

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
