## [Decision Letter · Decision Letter 0]

8 Jul 2021

PONE-D-20-38908

Multiple Criteria Optimization (MCO): a gene selection deterministic tool in RStudio

PLOS ONE

Dear Dr. Cabrera-Rios,

Thank you for submitting your manuscript to PLOS ONE. After careful consideration, we feel that it has merit but does not fully meet PLOS ONE’s publication criteria as it currently stands. Therefore, we invite you to submit a revised version of the manuscript that addresses the points raised during the review process.

According to referees' suggestions (see detailed comments below), some methodological approaches have to be better explained, some new results could be considered to be included (as gene enrichment analyses) and, in general, the format (typos and pictures) also improved.

We look forward to receiving your revised manuscript.

Kind regards,

Francisco J. Esteban, Ph.D., M.Sc.

Academic Editor

PLOS ONE

Journal Requirements:

Reviewers' comments:

Reviewer's Responses to Questions

**Comments to the Author**

1. Is the manuscript technically sound, and do the data support the conclusions?

Reviewer #1: Partly

Reviewer #2: Yes

Reviewer #3: Partly

2. Has the statistical analysis been performed appropriately and rigorously? 

Reviewer #1: Yes

Reviewer #2: No

Reviewer #3: N/A

3. Have the authors made all data underlying the findings in their manuscript fully available?

Reviewer #1: Yes

Reviewer #2: Yes

Reviewer #3: Yes

4. Is the manuscript presented in an intelligible fashion and written in standard English?

Reviewer #1: Yes

Reviewer #2: Yes

Reviewer #3: Yes

5. Review Comments to the Author

Reviewer #1: Gene selection is of great importance for the analysis of biological data. In this study, the authors proposed a method that select differential genes according to the Pareto criterion. Overall, the technical novelty is limited. Also, the authors claimed that they have used the proposed method “Our [22] previous works, Cruz-Rivera et al. [4], Lorenzo et al. [5], Isaza et al. [6], have 23 successfully developed and applied MCO for gene expression changes' selection in 24 conditions like Alzheimer's disease, cervix cancer, and lung cancer, respectively.”

There are points that help improve the manuscript.

1. How many datasets are used in the experiments and how are they used in this study? In Algorithm 1, the authors mentioned there are five datasets, while there are four Parkinson’s disease related datasets.

2. It is not clear how we get F? Also, rather than use 1, 2, 3, 4, and 5, it would be better use K to denote the number of datasets. Do rewrite Algorithm 1 to better illustrate it.

3. The pictures are of poor quality.

4. What are the differences between the MCO method with the commonly used filter, wrapper, embedded, and hybrid gene selection methods?

5. The discussion is kind of superficial, which needs writing.

Reviewer #2: 1. This work reported multiple criteria optimization (MCO) in gene selection for the analysis of microarray datasets. MCO selects genes with the largest expression changes without user manipulation of neither informatics nor statistical parameters. Furthermore, the user did not have to choose neither a preference structure among multiple measures of differential expression nor a predetermined quantity of genes to be deemed significant a priori. This implies that using the same datasets and performance measures (PMs), the method can converge to the same set of selected differentially expressed genes (repeatability) in spite of who the analyst is (objectivity). The reported work described the development of an open-source tool in RStudio to enable both: 1) individual analysis of single datasets with two or three PMs and 2) meta-analysis with up to five microarrays datasets, using one PM from each dataset.

2. In classification research, now I consider this contribution as pretty good to make a paper valuable for the field of interest. I am a convincing advocate of introducing the rigor of gene classification research in high dimensional datasets. However, I read this manuscript and I suggest to author please apply to show the competent with others recent statistical comparison method.

3. Quite good concept of paper and require complete language check should be performed to handle all typos and language issues.

4. There are many similar paper published on this topic, how your paper is different from existing ones? Explain.

5. Include some of the latest and relevant references for the benefit of the readers/authors of Cancer classification/ Parkinson disease based journal. The following citations will be very useful for the current, future and young research scholars in this research field from all over the globe.

a. Hybrid approach for gene selection and classification using filter and genetic algorithm.

b. Detecting biomarkers from microarray data using distributed correlation based gene selection

c. Dna gene expression analysis on diffuse large b-cell lymphoma (dlbcl) based on filter selection method with supervised classification method

d. An efficient stacking model with label selection for multi-label classification

e. Medical diagnosis of Parkinson disease driven by multiple preprocessing technique with Scarce Lee Silverman voice treatment data

f. Knowledge discovery in medical and biological datasets by integration of Relief-F and correlation feature selection techniques

The evaluation section is also not clear. Did authors use cross-validation or an independent test set? Did they train their model for one benchmark and used the trained model on the rest of the benchmarks?

Reviewer #3: The authors develop an open-source tool in RStudio for implementing their previous works about multiple criteria optimization (MCO) in gene selection for the analysis of microarray datasets. The novelty of this paper is not enough, and it needs further experiments. My suggestions are shown below.

1. In this paper and their previous works, they only used median or mean value to obtain differential genes. However, in many similar works about the selection of differential genes in two-class problems, researchers usually used more complicated rank-based measures, such as t-test, wilcoxon rank sum test, signal-to-noise ratio, etc. Authors should further demonstrate their MCO efficacy using such measures.

2. Please show their MCO can be further applied to multi-class problems.

3. Please further perform classification processes, i.e., selected features in conjunction with classifiers, to demonstrate their selected features are better than the selected features by other feature selection approaches for obtaining better classification results.

4. It needs to perform gene set enrichment analysis, such as DAVID. It can be used to show that the selected genes can be enriched in some biological processes or pathways.

6. PLOS authors have the option to publish the peer review history of their article (what does this mean?). If published, this will include your full peer review and any attached files.

Reviewer #1: No

Reviewer #2: No

Reviewer #3: No

---

## [Author Response · Author response to Decision Letter 0]

17 Nov 2021

Three reviewers provided feedback on our manuscript. Reviewer 1 raised six issues, Reviewer 2 raised five issues, and Reviewer 3 four issues. We provide our follow-up to each of them in the following sections and express our gratitude to the reviewers for their insight. The complete document is provided under the name "Response to reviewers.pdf". Some information might have been lost in the transcription below. 

Reviewer 1, Issue 1

Gene selection is of great importance for the analysis of biological data. In this study, the authors proposed a method that select differential genes according to the Pareto criterion. Overall, the technical novelty is limited. Also, the authors claimed that they have used the proposed method “Our [22] previous works, Cruz-Rivera et al. [4], Lorenzo et al. [5], Isaza et al. [6], have 23 successfully developed and applied MCO for gene expression changes' selection in 24 conditions like Alzheimer's disease, cervix cancer, and lung cancer, respectively.” There are points that help improve the manuscript:

Follow up to Reviewer 1, Issue 1

Thank you for your comment. The methods outlined in this paper have been, indeed, successfully tested on previous instances and their competitive performance has been demonstrated on previous works, as properly referenced in our manuscript. The resulting code presented here is an open-source scientific software, which fits the scope of The PLOS Computational Biology Software Section: 

(https://journals.plos.org/ploscompbiol/article?id=10.1371/journal.pcbi.1002799)

The code makes the analysis of single datasets efficient, reliable, and objective when pursuing gene selection. Furthermore, it allows the simultaneous meta-analysis of up to five datasets possible, which is a novel capability. This is especially true when considering that the datasets can be measured in different units, an issue that has majorly hampered the automation of meta-analysis. 

It is our belief that this code, as a computational artifact, brings strong analysis capabilities to gene selection that have not been completely available in the past, hence its novelty. 

Reviewer 1, Issue 2

How many datasets are used in the experiments and how are they used in this study? In Algorithm 1, the authors mentioned there are five datasets, while there are four Parkinson’s disease related datasets.

Follow up to Reviewer 1, Issue 2

We consider that this comment has to do with mixing up the capabilities of the MCO method, with those of the R code presented here, and not differentiating it from the number of datasets used in the illustrative example related to PD. We apologize for this confusion and proceed to disambiguate:

MCO is not mathematically/theoretically bounded in the number of datasets (and their corresponding performance measures) that it can analyze simultaneously. However, practical limits will arise related to the available computing power as well as the number of independent data points. 

The focus of the manuscript is the presentation of the automated R code for MCO, which can support meta-analysis of up to five databases simultaneously. So, even if MCO is mathematically unbound in the number of datasets that it can accommodate, the R code has been designed for up to five datasets. 

Finally, in the cases outlined for the illustrative example, four databases are available as described in Table 1, selected as explained in the Results section. These four datasets are used to support the analysis and meta-analysis cases detailed on Tables 2 and 3. 

To make sure that the distinction is clear, the material was organized in subsections: MCO algorithm, MCO tool, and MCO tool – Application, where the particular bounds are discussed. 

Reviewer 1, Issue 3

It is not clear how we get F? Also, rather than use 1, 2, 3, 4, and 5, it would be better use K to denote the number of datasets. Do rewrite Algorithm 1 to better illustrate it.

Follow up to Reviewer 1, Issue 3

Thanks for the advice. Besides the disambiguation provided previously, we have adopted the letter K to denote the number of databases of interest to the user to be analyzed through MCO (See Algorithm 1). For the R code, K falls between 1 and 5, and for the particular cases outlined in Tables 2 and 3, K is 1, 3, or 4 depending on the case. 

On the other hand, F is the only algorithm-related parameter that is user-defined. If F=1, the Pareto Frontier is identified with all data points involved in the analysis. If F=2, the first frontier is calculated with all data points involved and then the solutions to this first frontier are removed to proceed to a second run of the algorithm. This procedure is necessary to accommodate experimental error in the datasets. Obtaining more than 1 efficient frontier results in a hierarchy of solutions. 

In our experience, MCO converges to tens of solutions when analyzing tens of thousands initial genes. The number of frontiers (F) used on each analysis case in our manuscript is duly specified for single datasets (10 frontiers) and for meta-analysis (1 frontier). 

Reviewer 1, Issue 4

The pictures are of poor quality.

Follow up to Reviewer 1, Issue 4

We have tested our figures using the software provided by PLOS. All of them passed the quality inspection. In attention to the reviewer’s comment, we have increased the quality of our pictures even beyond these tests. The improved compliant set is now included in our resubmission.

In particular, we adjusted figures 1,3, 4,5 and 6. We also improved the following: the content in Fig 1 has been reorganized for clarity, the letter size was made larger for all graphs in the manuscript, and the size and letter type of the table in Fig 6 was modified to make it more readable. All figures passed the testing provided by the PACE program, as requested by the Journal instructions. 

Reviewer 1, Issue 5

What are the differences between the MCO method with the commonly used filter, wrapper, embedded, and hybrid gene selection methods?

Follow up to Reviewer 1, Issue 5

MCO is based on mathematical optimization. Mathematical optimization is applied when searching for the demonstrably best possible solution(s) to a particular problem. 

The demonstrably best possible solution to an optimization problem is called a global optimum, and it might or might not be accessible or recognizable in general. 

In our paper and our previous work, gene selection is casted as a multiple criteria optimization problem. In addition, our algorithm guarantees the global optimality of the solutions (that is, the resulting selected set of genes) thanks to the Pareto efficiency conditions. 

So, MCO guarantees global optimality, which -to the best extent of our knowledge- is a characteristic not afforded by the other methods. 

In addition, MCO arrives to the globally optimal solutions to the gene selection problem without asking the user to define a priori a final number of solutions, a preference structure between the different performance measures, the value of a threshold, or the use of a particular normalization procedure. The vast majority of gene selection methods would have at least one of these requirements. The fact that MCO does not require any of them, makes our method objective and repeatable. 

Finally, as demonstrated by the cases outlined in Table 3, MCO can select important genes based on the simultaneous consideration of multiple datasets, in spite of the underlying experiments being incommensurable (i.e. measured in dissimilar units). This is, indeed, a capability unique to MCO. 

Reviewer 1, Issue 6

The discussion is kind of superficial, which needs writing

Follow up to Reviewer 1, Issue 6

We are deepening biological interpretation based on Gene Ontology analysis in the Discussion Section to further support the biological significance of the results that are possible through the capabilities afforded by our method. 

Reviewer 2, Issue 1

In classification research, now I consider this contribution as pretty good to make a paper valuable for the field of interest. I am a convincing advocate of introducing the rigor of gene classification research in high dimensional datasets. However, I read this manuscript and I suggest to author please apply to show the competent with others recent statistical comparison method.

Follow up to Reviewer 2, Issue 1

When it comes to approaching a general class of optimization problems, such as those falling within Gene Selection, it is necessary to keep in mind the no-free-lunch theorem (NFLT) which states “that a general-purpose, universal optimization strategy is impossible. The only way one strategy can outperform another is if it is specialized to the structure of the specific problem under consideration”. In the development of MCO as presented in our manuscript, two important characteristics were considered: (1) the strategy should not require the manipulation of parameters for it to result in a competitive solution, and (2) the resulting solution should be globally optimal in presence of conflicting performance measures. The first characteristic would bring repeatability and the second one, objectivity. 

In Reviewer 1’s issues 1 and 5, we outlined and explained the distinctive advantages of our Gene Selection method, namely certainty (optimality), objectivity, consistent convergence, non-parametric nature, and the possibility of truly carrying out simultaneous meta-analysis in spite of incommensurability. These characteristics have been also discussed in our preceding publications. 

A comparative study is now available as supplementary material in attention to Reviewer 2’s Issue 1 and issue 3. This comparison includes four methods of gene selection: MCO, CFS, IG, and eBayes. The four methods were applied to the four datasets utilized in this manuscript. MCO was applied standardly with 10 frontiers as advised here, while the other methods were varied in their adjustment parameters. In addition, four types of classifiers were used to assess their classification accuracy: support vector machine, KNN, Treebag, Linear Discriminant Analysis and RF. 

The comparison involves classification accuracy, classification accuracy variance, and number of selected genes for the four gene selection methods across the datasets, the variation created by their parameters, and the classifiers. The results are as follows:

In terms of classification accuracy (Figure A1) , an ANOVA deemed that at least one of the methods had a different classification accuracy mean (p-value = 0.003). Furthermore, using Tukey´s multiple comparison scheme, it was determined that the CFS method provided the largest classification accuracy, while the other three (MCO, IG and eBayes) showed no significant difference between them at an alpha value of 0.05. This provides evidence of the competitiveness of MCO with the distinctive characteristic that no parameters need to be adjusted by the user. 

Regarding classification accuracy variance (dispersion), there was evidence that at least one of the methods had a different variance than the rest (p-values of 0.012 and 0.090 for Multiple Comparison´s and Levene´s methods respectively). Furthermore, through the analysis of the confidence intervals, it was determined that MCO´s interval did not overlap with the other three method’s intervals and that MCO´s interval contained lower values than the rest of the methods. This evidences MCO´s robust classification performance across different classifiers and datasets. 

Finally, regarding the number of selected genes (parsimony), an ANOVA showed that at least one of the methods had a different mean number of selected genes (p-value = 0.011). Furthermore, using Tukey´s multiple comparison scheme, two methods were deemed statistically equivalent in terms of their number of selected genes: MCO, CFS, both of which showed a significantly lower mean number of selected genes than eBayes. The IG method did not show significant difference with CFS and eBayes due to its large dispersion. This result evidences the parsimonious gene selection of MCO. 

So, in conclusion, this comparison shows the robustness of the solutions of MCO, the results of which tend to be parsimonious, providing a competitive classification accuracy performance without requiring the adjustment of statistical or computational parameters and with the unique capability to support simultaneous meta-analysis of multiple datasets. 

Figure A1. ANOVA and multiple comparison results for classification accuracy of the four gene selection methods. 

Figure A2. Test for equal variances of classification accuracy for the four gene selection methods. 

Figure A3. ANOVA and multiple comparison results for number of selected genes for the four gene selection methods. 

Reviewer 2, Issue 2

Quite good concept of paper and require complete language check should be performed to handle all typos and language issues.

Follow up to Reviewer 2, Issue 2

We have thoroughly revised our use of language. We used the online tool “Grammarly.com” to improve our exposition in this new version. 

Reviewer 2, Issue 3

There are many similar papers published on this topic, how your paper is different from existing ones? Explain.

Follow up to Reviewer 2, Issue 3

The reviewer is kindly referred to our follow-up to Reviewer 1´s Issues 1 and 5, as well as Reviewer 2´s Issue 1.

Reviewer 2, Issue 4

Include some of the latest and relevant references for the benefit of the readers/authors of Cancer classification/ Parkinson disease based journal. The following citations will be very useful for the current, future and young research scholars in this research field from all over the globe.

a. Hybrid approach for gene selection and classification using filter and genetic algorithm.

b. Detecting biomarkers from microarray data using distributed correlation based gene selection

c. Dna gene expression analysis on diffuse large b-cell lymphoma (dlbcl) based on filter selection method with supervised classification method

d. An efficient stacking model with label selection for multi-label classification

e. Medical diagnosis of Parkinson disease driven by multiple preprocessing technique with Scarce Lee Silverman voice treatment data

f. Knowledge discovery in medical and biological datasets by integration of Relief-F and correlation feature selection techniques

Follow up to Reviewer 2, Issue 4

Thank you for your suggestions. We have added the most relevant ones in our literature review and references. 

Reviewer 2, Issue 5

The evaluation section is also not clear. Did authors use cross-validation or an independent test set? Did they train their model for one benchmark and used the trained model on the rest of the benchmarks?

Follow up to Reviewer 2, Issue 5

The idea for this work is to result in biological relevance, thus our validation method goes along the lines of the biological evidence that emerges with our selection of genes. To this end, the discussion section was enhanced in this revised version. Although the classification problem was not the goal of this work, a complete comparison of classification performance is now added as an appendix following Reviewer 2´s Issue 1. This includes a comparison of four gene selection methods across multiple datasets, classifiers, and parameter variation. The cross-validation specifics have been added to this section, as these were varied for all methods for comparison purposes. 

Reviewer 3, Issue 1

In this paper and their previous works, they only used median or mean value to obtain differential genes. However, in many similar works about the selection of differential genes in two-class problems, researchers usually used more complicated rank-based measures, such as t-test, wilcoxon rank sum test, signal-to-noise ratio, etc. Authors should further demonstrate their MCO efficacy using such measures.

Follow up to Reviewer 3, Issue 1

The existence of several types of PMs measures and using a single one of them after a normalization procedure to select genes has contributed to the problem of reproducibility in this task. This was one of the reasons that motivated the generation of MCO. The rationale to use the (absolute) difference of medians or the difference of means has had to do more with transparency and with keeping assumptions to a minimum in our analyses. 

In the R code, there is a possibility of using the (absolute) difference of means, medians, third quartiles, or the kth-percentiles to give analysts further possibilities. 

A revision of our code will include adding a column to the dataset with values from performance measures calculated by the analyst with the additional information if these values are to be either maximized or minimized. This will swiftly accommodate the instances described by the reviewer. 

As long as a performance measure´s values are numerical and subject to either maximization or minimization, it can be perfectly analyzed using MCO, regardless of its measuring units and its complexity. Also, the possibility of carrying out meta-analysis (to up to five datasets simultaneously) remains unchanged in our code. 

Reviewer 3, Issue 2

Please show their MCO can be further applied to multi-class problems.

Follow up to Reviewer 3, Issue 2

MCO, as coded in R, can be applied to comparative experiments involving case (treatment) vs. control subjects. This is established in the Introduction section of the paper. 

MCO can meta-analyze multiple datasets concurrently provided that these datasets have a case-control layout as described previously. 

An application to multi-class problems is left for future work. 

Reviewer 3, Issue 3

Please further perform classification processes, i.e., selected features in conjunction with classifiers, to demonstrate their selected features are better than the selected features by other feature selection approaches for obtaining better classification results.

Follow up to Reviewer 3, Issue 3

The reviewer is kindly referred to Reviewer 2, Issue 1 which addresses this comment and provides a comparative study as suggested. 

Reviewer 3, Issue 4

It needs to perform gene set enrichment analysis, such as DAVID. It can be used to show that the selected genes can be enriched in some biological processes or pathways.

Follow up to Reviewer 3, Issue 4

This analysis is now provided in the Discussion section of the paper. Thank you for the suggestion.

---

## [Decision Letter · Decision Letter 1]

10 Jan 2022

Multiple Criteria Optimization (MCO): a gene selection deterministic tool in RStudio

PONE-D-20-38908R1

Dear Dr. Cabrera-Rios,

We’re pleased to inform you that your manuscript has been judged scientifically suitable for publication and will be formally accepted for publication once it meets all outstanding technical requirements.

Kind regards,

Francisco J. Esteban, Ph.D., M.Sc.

Academic Editor

PLOS ONE

Additional Editor Comments (optional):

Reviewers' comments:

Reviewer's Responses to Questions

**Comments to the Author**

1. If the authors have adequately addressed your comments raised in a previous round of review and you feel that this manuscript is now acceptable for publication, you may indicate that here to bypass the “Comments to the Author” section, enter your conflict of interest statement in the “Confidential to Editor” section, and submit your "Accept" recommendation.

Reviewer #1: All comments have been addressed

Reviewer #2: (No Response)

2. Is the manuscript technically sound, and do the data support the conclusions?

Reviewer #1: Yes

Reviewer #2: Yes

3. Has the statistical analysis been performed appropriately and rigorously? 

Reviewer #1: Yes

Reviewer #2: Yes

4. Have the authors made all data underlying the findings in their manuscript fully available?

Reviewer #1: Yes

Reviewer #2: Yes

5. Is the manuscript presented in an intelligible fashion and written in standard English?

Reviewer #1: Yes

Reviewer #2: Yes

6. Review Comments to the Author

Reviewer #1: The authors have addressed my concerns and the manuscirpt has been improved a lot. Please make sure that the references are closely related to the paper.

Reviewer #2: Good revised paper and well-written. Multiple Criteria Optimization (MCO): a gene selection deterministic tool in RStudio can be Accept.

7. PLOS authors have the option to publish the peer review history of their article (what does this mean?). If published, this will include your full peer review and any attached files.

Reviewer #1: No

Reviewer #2: No

---

## [Editor Report · Acceptance letter]

14 Jan 2022

PONE-D-20-38908R1 

Multiple Criteria Optimization (MCO): a gene selection deterministic tool in RStudio 

Dear Dr. Cabrera-Ríos:

I'm pleased to inform you that your manuscript has been deemed suitable for publication in PLOS ONE. Congratulations! Your manuscript is now with our production department. 

Kind regards, 

on behalf of

Dr. Francisco J. Esteban 

Academic Editor

PLOS ONE